# Impedance Matching and the Choice Between Alternative Pathways for the Origin of Genetic Coding

**DOI:** 10.3390/ijms21197392

**Published:** 2020-10-07

**Authors:** Peter R. Wills, Charles W. Carter

**Affiliations:** 1Department of Physics and Te Ao Marama Centre for Fundamental Inquiry, University of Auckland, PB 92019, Auckland 1142, New Zealand; 2Department of Biochemistry and Biophysics, University of North Carolina at Chapel Hill, Chapel Hill, NC 27599-7260, USA

**Keywords:** assignment catalysis, aminoacyl-tRNA synthetases, standard genetic code, dissipative processes, NTP hydrolysis, free energy transduction

## Abstract

We recently observed that errors in gene replication and translation could be seen qualitatively to behave analogously to the impedances in acoustical and electronic energy transducing systems. We develop here quantitative relationships necessary to confirm that analogy and to place it into the context of the minimization of dissipative losses of both chemical free energy and information. The formal developments include expressions for the information transferred from a template to a new polymer, I_σ_; an impedance parameter, Z; and an effective alphabet size, n^eff^; all of which have non-linear dependences on the fidelity parameter, q, and the alphabet size, n. Surfaces of these functions over the {n,q} plane reveal key new insights into the origin of coding. Our conclusion is that the emergence and evolutionary refinement of information transfer in biology follow principles previously identified to govern physical energy flows, strengthening analogies (i) between chemical self-organization and biological natural selection, and (ii) between the course of evolutionary trajectories and the most probable pathways for time-dependent transitions in physics. Matching the informational impedance of translation to the four-letter alphabet of genes uncovers a pivotal role for the redundancy of triplet codons in preserving as much intrinsic genetic information as possible, especially in early stages when the coding alphabet size was small.

## 1. Introduction

One of the impenetrable questions about biological evolution is how and why it is “open ended”. On geological time scales, systems of ever-increasing complexity appear and open up completely new possibilities of function, interaction and organized system structure. Nowhere is this more evident than at the origin of life, when functional molecular inheritance and translation systems, whose symbolic relationships could be used to write and read their own self-description, emerged from initial, high-temperature chemical disorder, presumably rather rapidly. How and why did it happen? Was it exclusively the outcome of some especially unusual configuration in the aqueous environment of the Earth’s surface? Or was it also driven by some innate tendency of far-from-equilibrium systems to create complex, spatially-differentiated dynamic states? How did the coded, functionally organized arrangement, typical of the nanoscale dynamic structure of living cells, first maintain itself and stave off the disrupting effects of thermal molecular activity?

The broadest answer to these questions, implicit in Schrödinger’s idea of a genetic “codescript”, is that enclosed microscopic systems somehow learned to compute; that is, they learned how to control their internal operations by embedding *symbolic information* into an “aperiodic crystal”, prophetic of the arbitrary sequences of linear nucleic acid heteropolymers found in the kernel of genetic programming. An important problem associated with this embedded information is that the quantity of it renders the probability of its undirected assembly very close to zero [1], unless it followed a stepwise, cooperative, and feedback-driven trajectory [2] in which the accumulation and maintenance of information at each step (i) mitigated the disordering effects of natural fluctuations in the requisite processes, and (ii) facilitated the simultaneous occurrence of whatever related events were able to set off the next transition. Recent results now facilitate serious investigation of these issues [2,3,4,5].

Analyses of the origin of genetic coding [2,5] recall Pauling’s prescient observation [6] that an important barrier to the emergence of translation was the limited ability of the earliest genetic codes to discriminate between similar amino acids. The presence of large editing domains in those aminoacyl-tRNA synthetases (aaRS) specific for amino acids from sets that are difficult to distinguish from one another argues persuasively that a minimal coding repertoire would inevitably have limited the specific recognition capabilities of the resulting aaRS. Even the rather advanced evolutionary models of Class I and II aaRS (Urzymes; [7]) select amino acids from the correct group of ten only 80% of the time [8,9]. Error rates of the earliest coding systems must have been very high. Further, as we have shown [3], the differential equations for the first systems of gene replication (and transcription) are strongly coupled through their respective population variables to those governing translation as a result of genes and gene products being made from different polymers, that is, because of the intrinsic separation of genotype and phenotype.

Connections between dissipation and the coupling of non-equilibrium systems to the free energy resources that drive them led to our hypothesizing that impedance matching should apply to the transfer of polymer sequence information and play an important role in determining the most probable pathways taken by the progressive self-organization and evolution of genetic coding [2].

In this paper, we investigate the dynamics and efficiency of information transfer and processing in prebiotic systems by examining how the mechanisms of nucleic acid replication (inheritance) and protein synthesis (translation)—copying and read/writing—were coupled during the evolution of genetic coding [10]. To that end, we develop a quantitative treatment of the impedance matching analogy between the optimization of the temporal rate of energy transfer in a physical system and the energy cost optimization of information transfer between biological polymers, especially the fundamental transfers specified in Crick’s Central Dogma [11]. The essentials of this work include both conceptual and experimental insights:

(i) Because translation is a computational manipulation of information independent of time, it is useful to formulate Ohm’s law in terms of energy cost, rather than the conventional elapsed time.

(ii) The formal analogy leads to quantitative expressions for I_σ_, the information transferred from a template to a polymeric product per monomer concatenated; and an impedance parameter, Z, constructed in terms of the coding alphabet size, n, and fidelity, q, giving a unified treatment suitable for describing both replication (information storage) and translation (information use).

(iii) Reformulating the terms “idling” and “stalling”, defined for electrical power transfer, in informational terms supports a *reductio ad absurdum* argument that impedance matching must lead to coordinated optimization of the relative error rates of replication, transcription, and translation.

(iv) Both replication and translation are subject to errors in transmission that reduce the effective alphabet sizes of the information they deliver. The resulting effective alphabet sizes can be computed on the basis of the error frequencies and, in the case of translation, the effect of code redundancy.

(v) The surfaces, I_σ_ and Z, define a landscape on which genetic coding developed and furnish a context in which to view the experimental properties of evolutionary intermediates of aaRS. The comparisons lend substantial qualitative support for our conclusions.

(vi) A surprising conclusion is that, as the coding alphabet dimension, n, increases, codon redundancy decreases the effective impedance, z, of the genetic template, buffering the lower coding error rate. That effect functions like a bicycle derailleur, accounting for much of the associated impedance matching between replication and translation along the gradual ascent over the I_σ_ and Z surfaces [2].

## 2. Results

In order to apply the impedance matching concept to biology, we first review its use in physics and especially in engineering, where it has significant practical importance in optimizing energy transfer [12]. We show that three types of dissipative losses in information have a similar formal structure to dissipative energy losses in engineering, and that in the course of building the standard genetic code, information dissipation was converted, via evolutionary changes, into dissipative losses in chemical free energy, leading to enhanced fidelity. Thus, in biology, errors in information transfer represent a form of impedance. Codon redundancy appears to have played a key role in assuring the stability of the first coded proteins.

### 2.1. Impedance Matching Optimizes Transfer of a Resource from a Driver to a Load

The most familiar example of impedance matching concerns transfer of electrical energy from a battery or generator, the “driver”, as a result of a “flow” (the current, I) through an electrical device that consumes the energy and is called a “load”. A driver can also be a load when it has energy transferred to it: for example, when mechanical or thermal energy is converted to electrical energy by driving a generator. In either case, when the impedances of the driver and load are the same, the power delivered, i.e., the rate of transfer of energy between them in either direction, is maximal. Impedance matching [13] also has important applications as an engineering principle in cases involving energy transduction, such as acoustics, where fluctuations in electric field are converted to sound waves, and hydraulics, where a pressure difference produces flow of fluid through a pipe.

The formal definition of a real scalar impedance is the rate of change of effort with respect to change of flow in the system when operating in a steady state. For a direct current circuit obeying Ohm’s law, V = IR, the “effort” is the voltage, V, that must be applied to the circuit to produce a particular current and the proportionality constant is the resistance, R, that impedes the flow of electricity in the circuit and effectively “consumes” the energy delivered to it. However, the potential difference across the terminals of a battery typically decreases from its rated value as the flow of current in the circuit (load) increases. This is due to the battery’s own internal resistance or “impedance” r. A simple textbook calculation [14] shows that, for a driver capable of a fixed electromotive force, E, and with a fixed (internal) impedance, r, the maximum rate of energy transfer from the battery to the circuit (power P = VI) occurs when the impedance of the load matches that of the driver, R = r, which illustrates the more general principle that impedance matching maximizes the rate of resource transfer to the load. Drivers are usually built so that their impedance can be varied to match different operational loads, allowing the rate of resource transfer to be maximized across a range of circumstances.

The current, I, is the rate at which electrical charge passes through any point in the circuit, measured in Coulombs per second (see Appendix A). For a DC circuit, the electromotive force E delivered by the battery is the difference in electrical energy held by the charges located on opposite terminals of the battery, measured in Joules per Coulomb. The units of electrical impedance (resistance, R) are Joule-seconds per Coulomb squared, which are called Ohms. Ohm’s law states the proportionality of the potential difference across a circuit to the current flowing through it: V = IR.

The purpose of impedance matching is to maximize the rate of energy transfer to the load. As charges move through the load circuit, they give up energy to it at a rate P = VI = I^2^R; for a circuit comprising only a simple resistor, all of the transferred is converted to heat. In this context, where the resistor is the “load”, the form into which the energy is converted is not the main point. What the “load” does with the energy transferred is no business of the “driver”. “Dissipation” in the driver/load context is reserved for the energy lost in the transfer process.

Likewise, energy is wasted at a rate P′ = I^2^r as a result of the impedance r of the source because the potential difference V across the circuit, i.e., the “load”, is less than the electromotive force E of the battery: E − V = Ir. However, as we will see (Section 2.4), when information flow is measured relative to the energy expended rather than the passage of time, maximizing the information flow corresponds to minimizing the energy dissipation driving it.

### 2.2. Informational Impedance Matching Likely Occurs in Molecular Biology

Impedance matching has recently been applied to the transfer of information [10], consistent with the idea of errors as an impedance to the process [15]. Section 2.3 and Section 2.4 derive the informational equivalent of Ohm’s law in order to provide a quantitative basis for elaborating the parallel application of impedance matching to the transfer of polymer sequence information in biology. The remainder of this section outlines relevant aspects of molecular biology that establish the need to consider how errors impact the flow of information. We distinguish the dissipation of chemical free energy from the dissipation of information, showing that the evolution of genetic coding effectively coopted the former in order to avert the latter as the number of distinct aaRS and their specificity increased (Figure 1); and we review the coupling between nucleic acid synthesis and translation (Figure 2).

The flow of energy in polymer synthesis is chemical, rather than electrical, and is driven by a chemical potential difference rather than a voltage. However, the “driver” of information flow is the information per monomer (or codon) in the template; the “load” is the polymer elaborated from the template, the destination of the information transferred. In translation, the load corresponds to the enzymatic aminoacylation of a tRNA followed by matching its anticodon to a corresponding codon in a messenger RNA. The chemical energy flow is the hydrolysis of ATP, which accomplishes information transfer from template to product, coupling information flow to energy flow. The essential enzymology entails a linear relationship between reaction velocity and total enzyme concentration (Equation (3) of [16]).

aaRS evolution entails two significantly different dissipative processes, as shown in Figure 1, both arising from activating the wrong amino acid. Contemporary aaRS have editing domains that dissipate chemical free energy by futile cycling through the ATP-dependent steps whenever a non-cognate amino acid is bound to the active site, dramatically increasing the energy dissipated to activate a non-cognate amino acid and the net energetic cost of faithfully activating cognate amino acids (Figure 1A; [17,18,19,20,21]). The aaRS equipped with error-correcting mechanisms are restricted to the subset that activate amino acids whose side chains, e.g., (Thr, Val), (Val, Ile), and (Leu, Ile), are similar enough that they cannot be selected without significant ambiguity by a single binding step. Experiments confirm that, when driven to acylate cognate tRNAs with the incorrect amino acid, these enzymes expend hundreds of ATP molecules for every incorrect amino acyl-tRNA formed [17]. That, normal, quality control process assures a low output error frequency.

In earlier stages of aaRS evolution (Figure 1B), significant numbers of misacylated tRNAs are incorporated into proteins, corrupting the translated products. Inherited information passed to the translation system by transcription is dissipated whenever composite errors in a translated product decrease its functionality. aaRS diversification and evolution enhanced translation fidelity by making better quality control increasingly expensive.

The Gene-Replicase-Translatase (GRT) model [22,23,24] (Figure 2) describes a minimal implementation of an explicit separation of phenotype from genotype. We use this heuristic to consider the relation between the relative impacts of errors in translation (creation of coded protein polymers from activated amino acid monomers) driven by replication (creation of nucleic acid polymers from activated nucleoside monomers).

### 2.3. Errors Impede Polymeric Sequence Information Transfer

For symbols chosen from an alphabet of size n, the information potentially available in a template polymer is I_T_ ≤ log_2_n bits. Three factors reduce I_T_ below the specified Shannon limit: (i) the finite accuracy of monomer recognition (q < 1) gives rise to errors; (ii) the total information available in a template cannot be transferred when the transfer mapping contains redundancy due to reduction in the size of the alphabet of chemical symbols used in the product; (iii) the distribution of sequences available as templates from which information is transferred to newly synthesized polymers depends, in turn, on the fidelity of replication.

#### 2.3.1. Loss Due to Noise, I′_σ_

We begin with a simplified Michaelis–Menten description of an enzymic reaction that transfers polymeric sequence information, either by adding a single nucleotide to a growing nucleic acid chain, complementary to an extant template, as occurs in DNA (or RNA) replication or transcription; or by coding, as occurs when an aaRS charges a tRNA with an amino acid. In each case, the enzyme distinguishes between a monomer, M, representing the “correct” choice, C, from a chemical alphabet and other monomers D_i_ that represent “errors”. We designate the usual two parameters for the correct choice as K_M_^corr^ (Michaelis constant) and k_2_ (catalytic constant); and K_M_^err^ and k_4_ for each possible incorrect choice. All monomers are assumed to be present at the same concentration, noting that the choice of which monomer is “correct” at each replication step is dictated by the nucleotide present at a particular template sequence position; and at each translation step by amino acid and tRNA identity. The ribosomal mechanism transfers the coding choices already made by aaRS enzymes into protein sequences by Watson–Crick base pairing to mRNA. The latter is also an imperfect process but it is mathematically convenient to bundle the effects of codon-anticodon mismatches into the kinetic parameters k_2_/K_M_^corr^ and k_4_/K_M_^err^ (see below).

Assuming that the free enzyme concentration, [E], is not significantly smaller than the total, E_0_, we can write the rate of incorporation, V, of monomers into linear polymers P, nominally comprised of L monomers, as
V = −(1/L)d[M]/dt = d[P]/dt = V_0_^corr^ + (n − 1)V_0_^err^ = k_2_ [EC] + (n − 1) k_4_ [ED_i_] ={(k_2_/K_M_^corr^) + (n − 1) (k_4_/K_M_^err^)}E_0_ [M](1)
This kinetic analysis of template-dependent polymer synthesis is less sophisticated than that undertaken by Hofmeyr et al. [25], but suffices for current purposes, as our interest is in the relative rates at which monomers are correctly matched, according to coding rules and/or base complementarity, to the information-carrying template.

We can then measure the accuracy of information transfer in terms of a “specificity parameter”,
σ = (k_2_/K_M_^corr^ − k_4_/K_M_^err^)/(k_2_/K_M_^corr^)(2)
which gives the fractional rate at which the “correct” as opposed to each of the (n − 1) “erroneous” monomers are incorporated into the polymer. In these terms, the rate Equation (1) becomes:−(1/L)d[M]/dt = d[P]/dt = {1 + (n − 1)(1 − σ)}V_0_^corr^(3)
The probabilities of correct (q) and incorrect (ε) monomer incorporation are thus
q = 1/{1 + (n − 1)(1 − σ)};    ε = (n − 1) (1 − σ)/{1 + (n − 1)(1 − σ)}(4)
or, using ζ = (n − 1)(1 − σ) = ε/q, the ratio of incorrect to correct assignments, we have the simpler expressions
q = 1/(1 + ζ);   ε = ζ/(1 + ζ)(5)
When σ = 1 (ζ = 0, no errors), we have q = 1 and ε = 0; when σ = 0 (ζ = n − 1, random polymer synthesis, correct assignment having the same chance as any one of the incorrect assignments), we have q = 1/n and ε = (n − 1)/n.

Consider the Shannon information measure for some value of the ratio of incorrect to correct assignments, ζ. Making a “correct” choice (by reading a template symbol and using a 1:1 assignment rule) is the same as making a choice of just 1 out of n possibilities, so it corresponds to the transfer of log_2_n bits of Shannon information. The loss of information (per monomer added to the polymer) due to errors in transmission processes is given in terms of the probabilities p_i_ of each of the n monomers i being chosen:I′_σ_ = −Σ_n_p_i_*log_2_ p_i_ = −q*log_2_ q − (n − 1)[ε/(n − 1)]log_2_[ε/(n − 1)] = −q*log_2_ q − ε*log_2_[ε/(n − 1)](6)
For specificity σ, I′_σ_ represents the information lost per monomer in transmission from the template to a newly synthesized polymer via replication, transcription or translation. Information *transferred* from a single monomer (for either replication or translation) is, under these conditions:I_σ_ = I_T_ + q*log_2_q + ε*log_2_[ε/(n − 1)] = I_σ_ + I′_σ_ = I_T_ ≤ log_2_n(7)
where I_T_ is the template’s average information content per symbol.

#### 2.3.2. Loss Due to Redundancy, I′_red_

Embedding information into a polymer that serves as a template depends entirely on the existence of a mechanism for distinguishing between the chemical features of the various monomers that can occupy any sequence position. Nucleic acid replication arises from base pairing and is simpler than translation in this respect. Its standard monomer alphabet {A,C,G,U/T} is of size n = 4 (log_2_4 = 2 bits of entropic information per monomer) and every sequence position is independently recognized.

Translation relies on mechanisms that are highly dependent on the evolution of protein binding sites, and thus on the coding alphabet size. Base triplet codons give mRNA sequences a putative alphabet size n = 64 (log_2_64 = 6 bits per codon). Even so, the total information content of a nucleic acid sequence that is L bases long, (2 bits/base) × (L bases) = 2L bits, is the same when it is considered as a sequence of triplet codons, (6 bits/triplet) × (L/3 triplets). There is a slight complication with translation in that 3 of the triplets (amber and stop codons) do not map onto an amino acid in protein sequences, so the operational size of the codon alphabet for information transfer is n = 61, which dictates an information density of log_2_61 = 5.93 bits/codon for open reading frames. 

The redundancy of the Standard Genetic Code (SGC), a surjective mapping from an alphabet of size n_C_ = 61 base triplet codons onto an alphabet of size n_P_ = 20 amino acids, adds a further complication. In the hypothetical case that all amino acids had the same redundancy factor, *ρ* = n_T_/n_P_, translation would result in a reduction in the quantity of information, I′_red_ = log_2_(61/20) = 1.61 bits per codon relative to that maximally available in the template. In the case of non-uniform redundancies the reduction in template information is given by
I′_red_ = (1/61)Σ*ρ*_i_log_2_*ρ*_i_(8)
where *ρ*_i_ is the redundancy of the ith entry in the amino acid alphabet. Equation (8), which is exact for sequences in which all 61 codons occur with equal frequency, yields a value of I′_red_ = 1.79 bits per codon for the SGC. This is a good indicative estimate of the loss of entropic information due to coding redundancy in protein synthesis, even though amino acids do not occur in proteins in exact proportion to the redundancy of their codons and the genetic usage of redundant codons is non-uniform. By such approximate reckoning the amount of information that is transferred, when a mRNA template is translated into a protein sequence using the SGC, has, neglecting errors, an estimated mean value I_σ_ = log_2_n_T_ − I′_red_ = 5.93 − 1.79 = 4.14 bits/codon. Calculations based on representative values of actual amino acid frequencies in proteomes [26] put the information density in the protein sequences of viruses, archaea, bacteria and eukarya in the range 4.13 to 4.20 bits/monomer, confirming the validity of our approximations to modern molecular biological systems, which operate at very low error rates.

#### 2.3.3. Internal Impedance of the Driver

Equations (1)–(8) have been developed without any necessary specification of the polymeric form, whether it be DNA, RNA or protein, into which the information “read” from the template is “written”. The distribution of individual sequences within the cloud of mutant variants surrounding the “master sequence” of a classic quasi-species is well known [27], but for the current purposes Equation (6) suffices to specify the information dissipated due to errors in replication. In particular, Equation (6) applies to the replication of genetic information preserved (presumably) in RNA templates when its translation into peptide/protein form was at its most primitive stage. Errors in template replication produce a population of sequences, which, by comparison with one another, have an average information content per symbol that is less than log_2_n_T_. The introduction of this noise in template replication can be accounted for using Equation (6) and gives rise to an average loss of I′_T_ per symbol (monomer or codon).

The information dissipated in the error-prone replication of a nucleic acid quasi-species, I′_T_, can be calculated by considering an alphabet of size 4 and therefore setting n = 4 in Equation (5). This process produces a quasi-species distribution with an entropic information content of log_2_ n_T_ − I′_T_ = 2 − I′_T_ bits per nucleotide monomer or I_C_ = 6 − 3I′_T_ bits per base triplet codon (using n_C_ = 64) for the process of translation.

These considerations are summed up in the equation
I_σ_ = (log_2_n_T_ − I′_T_) − I′_red_ − I′_σ_(9)
As the information loss due to replication errors occurs prior to translation, the terms in parentheses are most sensibly accounted for as a reduction in the effective (codon) alphabet size of the information available for transfer from the template:log_2_n_C_^eff^ = log_2_n_C_ − I′_C_(10)
This in turn requires a modified assessment of the information reduction due to code redundancy to ensure that translation cannot transfer information that does not already exist in the template:I′_red_ = log_2_(n_C_^eff^/n_P_)(11)
where n_P_ is the amino acid alphabet size. This leads to the final result
I_σ_ = log_2_n_P_ − I′_σ_ = log_2_n_P_^eff^(12)
Note that I_σ_ has been recast as the logarithm of an effective monomer alphabet size, log_2_n_P_^eff^, (Figure 3) whence it is obvious that errors in information transfer and the alphabet size of the target-polymer are opposite sides of the same coin: increasing I′_σ_ is equivalent to decreasing log_2_n_P_^eff^ ≡ I_σ_. 

### 2.4. The Informational Analogy to Ohm’s Law Appears in the Energy Domain

If a set of polymers is synthesized by transferring information at a density of I_σ_ (bits per monomer), then the information transferred per generation (of a defined set of m distinct sequences of length L) is
I = mLI_σ_(13)
provided the transfer at each sequence position is independent of other positions. If we divide both sides of this expression by U, the energy expended to synthesize the system of polymers, we can establish (i) a connection between information transfer through the process of polymer synthesis and the thermodynamics of that process and (ii) an equivalence between Equation (13) and the Ohm’s law expression, P = IV, for the power P transferred to the load circuit as a result of a current I passing through the potential difference V across the load.

Loss of information parallels, in the energy domain, the voltage drop between driver and load in the time domain. The reference dimension against which the information transfer is measured is chosen to be the energy expended rather than elapsed time because the result of a properly specified computation is the same, irrespective of how fast the individual algorithmic steps are traversed. Different algorithms may produce the same result in different numbers of steps, a measure of the algorithmic efficiency, but in physical systems the magnitude of energy flows determines which pathways are selected. Further, time and energy are often recognized as conjugate variables, hence affording different reference frames for the same phenomenon [28]. Thus, if real-world devices operate at very different speeds, the energy cost per step provides a scale, all else being equal, against which efficiency can be measured. This is particularly true of biological systems, which, literally as a matter of life or death, are constrained by the means available to them to deploy limited energy resources economically. 

Table 1 summarizes equivalences that follow from derivations in Section 2.3. We designate: (i) I/U = η, the information transferred per unit of energy expended in polymer synthesis, as the analog of power; (ii) mL/U = μ, the number of monomers concatenated onto polymers per unit energy expended, as the analog of current; and (iii) I_σ_, the information per monomer transferred to the newly synthesized polymer as the analog of the potential difference, the change in the potential energy per unit charge passing through the load circuit. Equation (13) then takes the form
η = μI_σ_(14)
in exact analogy to P = IV.

The detailed analogy between electrical energy transfer and polymer sequence information transfer extends to how the respective processes are driven. In the electrical case, a battery with an electromotive force E produces a potential difference V across a load circuit of resistance R, through which a current I flows. In the case of template directed polymer synthesis processes, such as molecular biological replication, transcription and translation, a polymer template sequence with an average information content per symbol, I_T_, supplies information which is transferred in a stepwise fashion into a new polymer sequence as monomers are concatenated. Just as V ≤ E for a circuit connected to a battery, we require I_σ_ ≤ I_T_; and just as I(E − V) corresponds to a power loss due to the internal resistance r of the battery, so too is there a loss of information per unit of expended energy μ(I_T_ − I_σ_) due to noise (errors) in the transfer process. We elaborate explicitly on correspondences of units and dimensionality in Appendix A.

Extending the analogy to the elementary form of Ohm’s law, V = IR, we would expect to find I_σ_ = μΖ, that is, the (average) quantity of information transferred per monomer is proportional to the quantity of information per unit energy expended in polymer synthesis multiplied by an impedance parameter Z. Z is the ratio I_σ_/μ and as 1/μ is the energy cost of incorporating a monomer into the polymer, the impedance is the product of the information transferred per monomer concatenated to a growing polymer multiplied by the energy cost of monomer concatenation, irrespective of whether such an event transfers information. The transfer of information from a template into the sequence of a new polymer molecule is intrinsically harder when it is carried out either accurately (I_σ_ large) or when it has a high energy cost (μ small).

### 2.5. Variation of “Effort” with “Flow”

The simple linear form of Ohm’s law, V = IR, and its analog for polymer sequence information transfer, I_σ_ = μΖ, are valid only within certain ranges of drive “effort”, V or I_σ_, and the load “flow”, I or μ. On the other hand, the idea that the rate of resource transfer, P or η, is maximized when the load impedance, R or Z, is matched to the impedance of the driver, r or z, can be generalized to real cases in which the effort–flow relationship is markedly curvilinear. This is done by defining an operational impedance for any working state of a system as the differential of the effort with respect to the flow, R = dV/dI or Z = dI_σ_/dμ. The generalized impedance for the transfer of polymer sequence information is found by differentiating both I_σ_ and μ with respect to temperature to calculate dI_σ_/dμ as (dI_σ_/dT)/(dμ/dT), for which purpose we consider the effect of temperature on the activation energy and reaction rate of an enzyme catalyzed reaction. 

The temperature dependence of I_σ_ is readily evaluated from Equation (6) by first differentiating I_σ_ w.r.t. q to obtain
(15)$${\mathrm{dI}}_{\sigma }/\mathrm{dq}=\log_{2}q-\log_{2}\left[\left(1-q\right)/\left(n-1\right)\right]$$
and then finding the temperature dependence of q through its relationship to the kinetic parameters given by Equations (2) and (4). The logarithm of ratio k_cat_/K_M_ is the free energy of activation, −ΔG^‡^/RT, for the second-order enzymatic reaction of substrate and free enzyme to give product, whence we find
(16)$$\begin{array}{cc} \ln\lambda & =\ln\left(1-\sigma \right)=\ln\left[\left(k_{4}/K_{M}^{\mathrm{err}}\right)/\left(k_{2}/K_{M}^{\mathrm{corr}}\right)\right]\\ & =\left[\Delta G^{‡}(\mathrm{corr})-\Delta G^{‡}(\mathrm{err})\right]/\left(\mathrm{RT}\right) \end{array}$$
First, we use the identity q = 1/[1 + (n − 1)λ] to differentiate q w.r.t. λ. Second, ignoring any minor temperature dependence of the free energy of activation for the reaction, we use Equation (16) to differentiate λ w.r.t. T. Third the chain rule is applied to obtain
(17)$${\mathrm{dI}}_{\sigma }/\mathrm{dT}=\frac{-\left\{\log_{2}q-\log_{2}\left[\left(1-q\right)/\left(n-1\right)\right]\right\}\left(n-1\right)\lambda \ln\lambda }{{\left\{1+\left(n-1\right)\lambda \right\}}^{2}T}$$

The temperature dependence of μ is more straightforward. The energy cost, u = U/mυ = 1/μ, of concatenating another monomer M to a polymer P_l–1_ of length l–1 monomers depends on the free energy difference between the reactants and products of an enzymatically catalyzed chemical reaction of the form
M + P_l–1_ + pATP → P_l_ + pAMP + pPP_i_(18)
where we have assumed that the monomer addition reaction requires concomitant enzymatic hydrolysis of p nucleotide triphosphate units. The coupled catalysis of polymer synthesis and ATP hydrolysis is only possible under circumstances where this overall reaction is displaced from the equilibrium condition
(19)$$\Pi_{i}[{\mathrm{product}}_{i}{]}_{\mathrm{eq}}/\Pi_{i}[{\mathrm{reactant}}_{i}{]}_{\mathrm{eq}}=K_{\mathrm{eq}}$$
and then the free energy cost of the reaction can be measured as
(20)$$u=\mathrm{RT}\ln\left\{K_{\mathrm{eq}}\Pi_{i}\left[{\mathrm{reactant}}_{i}\right]/\Pi_{i}\left[{\mathrm{product}}_{i}\right]\right\}=1/\mu$$
and
(21)du/dT=−1/(uT)
This allows us to define the differential form of the impedance, dI_σ_/dμ, in terms of the information transfer specificity parameter, ζ, as
(22)$$Z=\frac{{\mathrm{dI}}_{\sigma }}{d\mu }=u\zeta {\left\{\frac{1}{\left(1+\zeta \right)}\log_{2}\left[\frac{\zeta }{\left(n-1\right)}\right]\right\}}^{2}$$
and the corresponding expression for the information transferred from the template is obtained from Equation (7) as
(23)$$I_{\sigma }=\log_{2}n_{P}-\log_{2}\left\{{\left[\zeta /\left(n-1\right)\right]}^{\zeta /\left(1+\zeta \right)}/{\left(1+\zeta \right)}^{1/\left(1+\zeta \right)}\right\}.$$

It is notable that when the impedance is defined as a differential it depends only on the dissipative part of I_σ_, that is on I′_σ_ = −Σ_n_p_i_*log_2_ p_i_ (Equation (6)), and not on log_2_n_P_ = log_2_n_P_^eff^ − I′_red_, the information available from the template, which depends on redundancy between the alphabets used for replication and translation, as well as information dissipation during replication.

The relative values of the impedance parameters, Z and z, are most easily interpreted in terms of the useful transfer (Z) and dissipative loss (z) during the process of polymer sequence information transfer during either replication or translation. These impedance parameters are influenced by three factors: the energetic cost, u, the error rate, ε, and the size of the monomer alphabet, n. 

### 2.6. Severe Impedance Mismatching Leads either to Idling or Stalling

We can now interpret the two extreme conditions that afflict information transfer in terms of the temperature dependence of the enzymatically effected process: (i) idling, when = n − 1, |ΔG^‡^(corr) − ΔG^‡^(err)|/RT → 0, i.e., T → ∞; and (ii) stalling, when = 0, |ΔG^‡^(corr) − ΔG^‡^(err)|/RT → ∞, i.e., T → 0. It does not matter that neither of these two extremes of temperature corresponds to physically realistic circumstances. Rather, what does matter is that in physically realistic circumstances the temperature dependence for the rate of enzymatically catalyzed reactions actually conforms to the manner in which it is represented, Equations (1), (16) and (20), which are standard biochemical expressions for the kinetics and thermodynamics of the process. 

In the electrical analogy, the extreme cases of impedance mismatching lead to the breakdown of energy transfer: (i) for R >> r with fixed E and r, I → 0, V → E, the circuit acts as an insulator; and (ii) for R << r, V → 0, there is an effective short circuit with power loss, P′, approaching IE due to the internal resistance r of the battery. In the case of polymer sequence information transfer: (i) for Z >> z, the process becomes perfect (q = 1, ε = 0, σ = 1, ζ = 0), μ → 0, I_σ_ → I_T_, but the cost of polymer synthesis becomes so high that the whole process stalls; and (ii) for Z << z, polymer synthesis is energetically cheap but it is random and no information is transferred (q = 1/n, ε = 1, σ = 0, ζ = n − 1), I_σ_ → 0.

The limiting cases of idling and stalling necessarily imply that successful information transfer requires at least some degree of matching between Z and z. It is easy to envisage polymer information transfer ending up in the idling state (random synthesis), but true stalling would require an infinite discrepancy between the activation energies for correct and erroneous information transfer. In practical engineering terms, finding operational conditions under which the driver impedance matches the load impedance serves to optimize the flow of the resource and avoid stalling or idling. However, failing to do so may allow the system to enter a regime of negative impedance and lead to device failure or destruction [12]. Likewise, the dissipation of information during transfer can be ascribed to the internal impedance z of the driving process: errors are inevitable. The Ohm’s law expressions for information transfer, loss and dissipation are I_σ_ = μZ, I′_σ_ = μz and η′ = μ^2^z, respectively, and the impedance-matching condition Z = z maximizes the transfer of information relative to the energetic efficiency of polymer synthesis, just as the rate of energy transfer P from a driving DC electromotive device to a purely resistive load circuit is a maximum when the driver and load are impedance matched, i.e., R = r [14].

### 2.7. Information Is Preserved at Thermodynamic Cost

A substantial body of relatively recent work has greatly clarified the connections between computation and the statistical thermodynamics of irreversible processes and enlightened the true cost of computation in real physical devices [29]. Modeling has revealed a linear dependence of log(ε/ε_0_) as a measure of performance in biology on the free energy cost [30]. The power dissipated in computation exceeds the minimal cost of erasing bits of information [31] by many orders of magnitude. Molecular biological systems are the most efficient known, but even they operate with inefficiencies at least 20 times and up to 10^4^ times the ideal minimum [32]. Our previous consideration of impedance matching considered only that limit [2] mirroring Schneider’s discussion of the cost of inserting a correct monomer into a growing polymer by means of a process that is computationally reversible [33].

The energy directly expended attaching an amino acid to a growing peptide chain is, to a first approximation and in the absence of specific editing, independent of whether it is correctly or erroneously matched to the mRNA codon coupled to the tRNA occupying the A site of the ribosome. Hydrolytic expenditure of both phosphates of ATP is required to attach an amino acid directly to a tRNA and expenditure of a further two high energy phosphates (as GTP) is required to transfer the amino acid to the nascent peptide at the ribosomal peptidyl transfer center, i.e., hydrolysis of a total of 4 ATP molecules [34]. (We introduce the additional cost of editing steps in Figure 1 and discuss it further below in Section 3.4.) Thus, any difference between the Shannon information content of genes and proteins arises from the differences in the relative frequencies of monomers, nucleotides in genes and amino acids in proteins. On the broadest scale, the sequence of an organism’s genome can be considered to be one chosen randomly from all possible sequences of length L made up of the four letters {A,C,G,T/U}, of which there are 4^L^, whence the entropic information content of nearly every sequence is very close to 2L bits. The only major modification of this estimate arises due to inequality in the proportion of GC and AT base-pairs in the average entropic information content of genomic nucleotide sequences. For bacteria, the genetic GC content can vary from 25% to 75% [35]. Given that these GC values are oppositely coupled to AT values of 75% and 25%, the number of bits of information per nucleotide for either of these extreme GC values can be estimated as −2 × {0.25 × log_2_(0.25) + 0.75 × log_2_(0.75)} ≈ 1.6, giving between 4.8 bits and the maximum 6 bits per nucleotide-triplet codon when all four nucleotides {A,C,G,T} have equal frequencies. A standard measure of the relative frequencies of amino acids in proteins [36] yields a value of I_σ_ = 4.2 bits per monomer, demonstrating that the simple entropic information loss due to the process of translation is in the range 0.6 to 1.8 bits per amino acid or 10–33% of the information in a codon. Only a miniscule fraction of the observed variation in the average monomer information content of organisms’ genes and proteins can be attributed to errors in translation, so these reductions are measures of the combined effect of coding redundancy and non-uniform amino acid usage in proteins. 

### 2.8. Impedance Matching

The detailed analogy between electrical energy transfer in a circuit with a simple resistive load and information transfer between polymer templates implies that the information transfer from genes to functional catalysts will be most cost-efficient when the processes of replication and translation both satisfy the same impedance-matching condition, because the replication “load” is the translation “driver”. We summarize in Section 3 the unexpected experimental support for our conclusions.

## 3. Discussion

The definitions of parameters developed in Section 2 (transfer rate, η, Equation (14); carrier flow, μ; potential, I_σ_, Equation (6); and impedance, Z, Equation (22); Table 1) furnish a comprehensive quantitative framework for analyzing non-linearities in the dynamics of transfer of polymer sequence information in replication and translation. Unexpected experimental support for that framework comes from the limited fidelity and apparent alphabet size of well-characterized models for an early, but important stage in aaRS evolution (Section 3.1). The functional dependences of I_σ_ and Z in the {q, n} plane place substantive constraints on the nature of the early stages of genetic coding (Section 3.2), underscoring our previous proposal that genetic coding advanced by successive quasi-species bifurcations that progressively increased the coding alphabet size [2]. Codon redundancy in the genetic templates for protein synthesis substantially reduced the impedance mismatch between replication and translation in early stages of coding and this contribution decreased automatically as the alphabet size increased (Section 3.3). Finally, the utility of the analogy between electrical power transmission and polymer sequence information flow suggests that biological processes may be governed by minimizing dispersion, just as physical processes are (Section 3.4).

### 3.1. Experimental Models for Ancestral aaRS Exhibit Properties Consistent with the Constraints Imposed by the {q, n} Dependencies of I_σ_ and Z

We have constructed and validated the full functionality (amino acid activation, single-turnover burst, and aminoacylation [37]) for two Class I Urzymes (TrpRS and LeuRS) and one Class II (HisRS). Much of the work on the LeuRS Urzyme (J. J. Hobson, Unpublished) has been published piecemeal [8] owing to intermittent funding and changes in personnel. The two Class I Urzymes behave equivalently in two key respects. First, the LeuRS Urzyme has significantly higher amino acid activation activity than the TrpRS Urzyme. Second, we extended the analysis of codon middle-base pairing [38] showing that TrpRS and TyrRS Urzymes had significantly elevated middle-base complementarity in alignments antiparallel to HisRS and ProRS Urzymes, initially including eight additional MSAs drawn from both Class I and Class II sequences (incl. LeuRS Urzyme), and then to all 20 aaRS. Elevated middle-base pairing (mbp) is found for antiparallel alignments for all 20 Urzyme sequences (<mbp>_overall_ = 0.34 ± 0.005). In particular, LeuRS Urzyme has a mean codon mbp frequency of 0.37 ± 0.01 against all Class II sequences in the sample. As this value is higher than that for TrpRS (0.34 ± 0.001), we consider LeuRS to be a better representative of Class I Urzymes.

The observed substrate specificities of reconstructed exemplars of Class I LeuRS and Class II HisRS Urzymes (Figure 4A) furnish non-trivial experimental validation for the I_σ_ and Z dependences. They have two relevant interpretations. The histogram in the center of Figure 4A shows that both Urzymes activate amino acids from within their own class with a free energy preference of ~1 kcal/mole. That value means that they administer a binary alphabet, n = 2, with a q value of 0.8. However, within-class discrimination between amino acids is higher. Class I LeuRS Urzyme recognizes five of the ten Class I amino acids with a mean ΔΔG(k_cat_/K_M_) = −2.8 kcal/mole corresponding to a q value of 0.99. Class II HisRS Urzyme recognizes five of the ten Class II amino acids with a mean ΔΔ(Gk_cat_/K_M_) = −1.3 kcal/mole, corresponding to a q value of 0.89. The two extant aaRS Urzymes therefore divide the 20 canonical amino acids roughly into four sets, each with five members, which is a key characteristic of the enzymatic administration of a four-letter alphabet. Efforts are underway to extend the analysis in Figure 4A to other Urzymes, and to constructing additional Urzymes using reduced amino acid alphabets, perhaps even imposing coding by a four-letter alphabet. Thus, these questions are now accessible to experimental analysis.

Figure 4B,C show plots of the potential, I_σ_, and relative impedance, Z/u, as functions of the accuracy of transfer, q, and the codon alphabet size n_C_. Visible parts of both surfaces show the landscape over which genetic coding evolved as fidelity and alphabet size increased. Increasing alphabet size along the red path fails to increase I_σ_ significantly and ultimately follows a steeper path over the ridge in Z/u and for that reason is considered less probable. The yellow paths increase q without a corresponding increase in n until sufficient specificity is achieved to benefit from an increased alphabet size, circumventing the steepest part of the ridge in Z/u. Remarkably, experimental data in A, shown by the star, are approximately consistent with the yellow path.

It is worth noting that the sharp initial clockwise turns marked by stars on the yellow curves in Figure 4B,C have additional support in the literature devoted to what is known about how aaRS achieved their high fidelity. Class I aaRS all have a variable insertion, connecting peptide 1 (CP1), between their N- and C-terminal halves, which interrupts the bidirectional coding and which contains the editing domains in the larger Class Ia aaRS [39,40]. We have discussed elsewhere [41,42,43,44,45] the evidence that CP1 sequences were acquired late in aaRS evolution, and that the CP1 and anticodon-binding domains in all Class I aaRS interact in a cooperative, allosteric mechanism that enhances amino acid specificity. These sophisticated mechanisms would have been necessary to lift the set of evolving aaRS along the ridge defined by the sequence of yellow points in Figure 4B,C, up the I_σ_ slope close to q = 1, enabling the corresponding increases in alphabet size.

### 3.2. Initiation and Stepwise Evolution of Coding

There is overwhelming evidence that the two modern aaRS superfamilies descended from small (M_W_ ~14 KD) Class I and II Urzyme-like proteins encoded by the +/− strands of a single gene [8,43,46,47,48]. Secondary structures of these two putative aaRS ancestors enabled them to discriminate between large (Class I) and small (Class II) amino acid side chains (Figure 4A; [4]), and between the reverse hairpin (Class I) and extended helical path (Class II) of the 3′-terminal DCCA extensions of available tRNA-like mini-helices [49,50]. These properties fulfill requirements for operating a binary (n = 2) code able to distinguish two subsets of the available amino acids as a minimal implementation of translation coupled to replication.

Persistence of a pair of Class I and II “founder” aaRS-type peptides would have required the concomitant preservation of the template information that they used in the mutually cooperative autocatalytic process of their own production. The dynamics of template-directed protein synthesis display inherent solutions to that problem as well as the emergence of coding [24,51,52]. Loss of genetic information from such systems as a result of noisy template replication can be mitigated by a very simple group selection mechanism. As has been demonstrated [23] in studies of GRT systems (Figure 2) that include proteins having RNA replicase-like activity, reaction–diffusion coupling produces the necessary grouping in Turing pattern dissipative structures without the need for protocell units that require stoichiometric reproduction of components. Thus, spontaneous chemical self-organization in a system supporting template-dependent protein synthesis is sufficient to account for the creation of a coding relationship between nucleic acid and protein sequences [51,53].

### 3.3. Decreases in I′_red_ with Increasing Alphabet Size Help Match Impedances

Modern organisms transfer information from nucleic acid genes into functional proteins with very high accuracy. Error rates per amino acid monomer, ε, are of order 10^−4^ or 10^−5^, sometimes even smaller [54]. Base pairing makes replication inherently accurate enough that the low cardinality of the nucleotide alphabet (n = 4) simplifies the problem of preserving the large body of template information required to code proteins with high functional specificity: any improvement in nucleotide discrimination has a close to threefold effect on the accuracy of codon replication. These operational circumstances mean that contemporary molecular biological translation can approach such a high fidelity that information flow can vary across its entire dynamic range with very little change of behavior (see dashed yellow arrow in Figure 4C, Section 3.1). Thus, the impedances, Z and z, are effectively similar to each other both at the origin of genetic coding and in contemporary biology.

Here, we outline how a succession of changes in the driver impedance and coding alphabet size likely eased the progression of the coding alphabet from an ancestral binary code to the SGC. Reduced translation errors coupled to decreases in replication errors enabled the dimension of the coding alphabet to increase, so coded protein enzymes became both more accurately translated and more diverse. That improvement, in turn, enhanced their functionalities, especially their specificities as the dimension of the coding alphabet increased.

Because replication and translation depend on the same energy source (hydrolysis of nucleotide triphosphates), maximizing the utility of the energy expenditure driving these joint processes subjects them to the same impedance-matching condition. Furthermore, although we may never know whether nucleotide triplets were employed as codon units from the very inception of binary coding, to assume otherwise unnecessarily complicates the task of understanding how the coding alphabet was able to expand to comprise the 20 canonical amino acids. Codon redundancy provided a further buffer against the difficulty of accurately reading codons, with the base-pairing mechanism further reducing the chemical complexity of that problem by another factor close to two. Thus, as corresponding increases in information per codon, I_σ_, and alphabet size, n, decreased the impedance, Z, of the load, decreases in I′_red_ by Equation (11) decreased the internal impedance, z, of the driver. As suggested previously [2], this compensation served a role analogous to the successive gears in a bicycle derailleur (Figure 5). This leaves scope for the recognition processes of replicative information transfer to evolve in parallel, eventually to error rates as low as 10^−6^ in prokaryotes and 10^−9^ in animals and plants.

Figure 5 provides a quantitative demonstration of the pathway of such an evolution in terms of the analogy, established here, between the flow of energy in physical systems and the flow of polymer sequence information in molecular biological systems.

Finer discrimination by more highly evolved aaRS required sophisticated allosteric behavior (Section 3.1) that could not be achieved until the size of the coding alphabet increased sufficiently. In this case, one solution was inventing a new, costly mechanism of computation in the form of kinetic proof-reading/editing (Figure 1). More generally, the progressive embedding of information into tRNA and genes via increases in the alphabet size necessarily also implies that proteins themselves must have gained functionality by the kind of stepwise progression envisioned by [55]. Were it not for the extremely high accuracy of nucleic acid replication and the high redundancy of the genetic code, the impedance mismatching implicit in this fine tuning of the genetic code would never have been possible.

Whatever aspects of the genetic code are purely “frozen accidents” [56], whatever aspects reflect stereochemical interactions between amino acids and codons [57], and whatever the order in which evolving metabolic processes could deliver different amino acids in quantities sufficient for their incorporation into proteins to play a significant role in prebiotic chemistry [58,59], the genetic code has evolved to be “regular” in the manner in which it assigns the chemical properties of amino acid sidechains to codons. In other words, chemical regularities in the code are an inevitable outcome of the stepped, bootstrapped, impedance-matched pathway that was available for its evolution (yellow dashed curves in Figure 4B,C) rather than the result solely of some kind of Darwinian search within the astronomically sized space of possible coding assignment patterns [1,60,61].

### 3.4. Uniting Biology with Physics

Impedance matching is used in engineering to maximize the rate of transfer of a resource, usually some useful form of energy. Transfer inevitably dissipates or loses some fraction of the resource via conversion to another form, often heat, with a corresponding overall increase in entropy, consonant with the second law of thermodynamics. However, energy transfer processes are time dependent and typically non-linear. Thus, they lie outside the realm of equilibrium thermodynamics, making them more challenging to analyze correctly [62,63].

Equations (22) and (23) provide a straightforward context and evolutionary solution to the vexing problem entailed in Eigen’s paradox [64]: maintaining more information in a nucleic acid template requires higher replication fidelity, which can only be achieved with the help of enzymes that function with increased specificity; but the encoding of increased specificity necessitates the prior preservation of more information. Eigen’s paradox is a specific instance of the more general chicken-egg problem: only those changes to the code that improve its computational faculties in a self-referential (reflexive) manner become entrenched. Proxy letters of an encoding alphabet always interact functionally with amino acid sidechains at aaRS active sites according to some scale of operational similarity. Thus, cause and effect coincide as a result of the feedback between them.

Physicists and physical chemists alike have developed approaches to the description of non-equilibrium thermodynamics [65] and Prigogine [66] has drawn special attention to the fundamental role of far-from-equilibrium processes in biology. Likewise, a close connection between the second law of thermodynamics, the principle of least action, and Darwinian natural selection has been suggested [67]. One approach that has dealt successfully with non-equilibrium processes, is the path integral of Lagrangian functionals [68]. Relationships connecting the minimum dissipation principle to variational treatments of stationary behavior were recognized by Onsager [62] and have since been shown to be much more general [63,69]. Indeed, the principle of least action leads to consistent conclusions throughout physics [70,71,72,73] and, recently, in bioenergetics [74,75]. The detailed mathematical analogy articulated in Section 2 and Table 1 between the evolution of genetic coding and the optimization of power transfer in physics argues that biology can be seen also to exhibit evolutionary paths in which dissipation in one domain (energy) ends up being harnessed to sustain order in another domain (information).

## 4. Materials and Methods

This work rests entirely on the novel theoretical derivations in Section 2, Section 3 and Section 4. No new experiments were performed; there are no methods to describe; no new computer programs were written; derivations in Section 2, Section 3 and Section 4 are presented therein in their entirety. Figures were prepared using Canvas [76] from plots made with JMP [77] (Figure 3 and Figure 4).

## 5. Conclusions

We have derived a formal equivalence between Ohm’s law and the relationship between the information content of a polymer and the frequency of errors accompanying its synthesis. The differential form of the derivation, Section 2.7, and the consistency of the experimental fidelity of aaRS Urzymes with the more likely path to complete the coding table, Figure 4, solidify our previous suggestion that increases in the numbers of letters in the coding alphabet served to match the error frequencies in replication and translation. That conclusion strengthens the case that the genetic coding table arose by a series of quasi-species bifurcations that enlarged the coding alphabet from an initial value of 2 letters to its current state of 20 letters. The dependence of the informational impedance, Z, on the error rate, q; and the alphabet size, n, from the Ohm’s law equivalence shows that the coding table had to cross a ridge for which the lowest barrier is at low n. The most probable path increased the fidelity of a limited coding alphabet also provided the most gradual, continuous increase in the genetic information per monomer, I_σ_. The high experimental error frequency and low apparent alphabet size observed for aaRS Urzymes are consistent with that subset of paths. Thus, after each increase in the size of the alphabet, the extant aaRS must have optimized the associated error frequency, as they took advantage of newly created coding diversity to increase their specificity from a very low value at the onset of translation. The full contemporary genetic alphabet represented by the contemporary codon table was achieved in large part by making non-cognate catalytic assignments by aaRS increasingly energetically costly, generating futile cycling of ATP hydrolysis, thereby converting informational dissipation into energetic dissipation. The mechanistic sophistication of those mechanisms themselves required a much higher value of n. Finally, progressive decreases in I′_red_ confirm our previous suggestion that increases in the numbers of letters in the coding alphabet served to match the error frequencies of replication and translation.

## Figures and Tables

**Figure 1 ijms-21-07392-f001:**
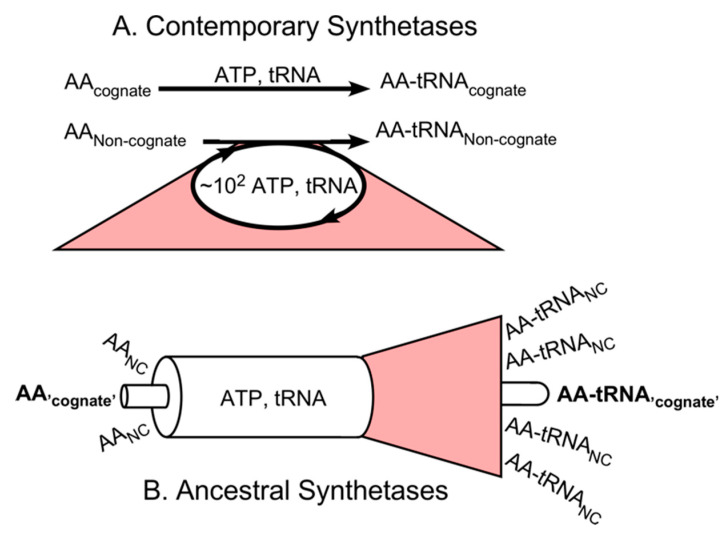
Dissipation of chemical free energy and information by aminoacyl-tRNA synthetases. (**A**) Contemporary aaRS are highly specific. When stereochemical differences between similar amino acids are small enough to permit ambiguity in a single binding event, specificity is enhanced by futile cycling of ATP consumption in the presence of non-cognate amino acids, dissipating chemical free energy. (**B**) Ancestral synthetases were very likely sufficiently less specific to allow incorporation into translated peptides of non-cognate amino acids at levels that led to the corruption of a large percentage of the translated peptides, leading to dissipation of the information embedded in the mRNA codescripts.

**Figure 2 ijms-21-07392-f002:**
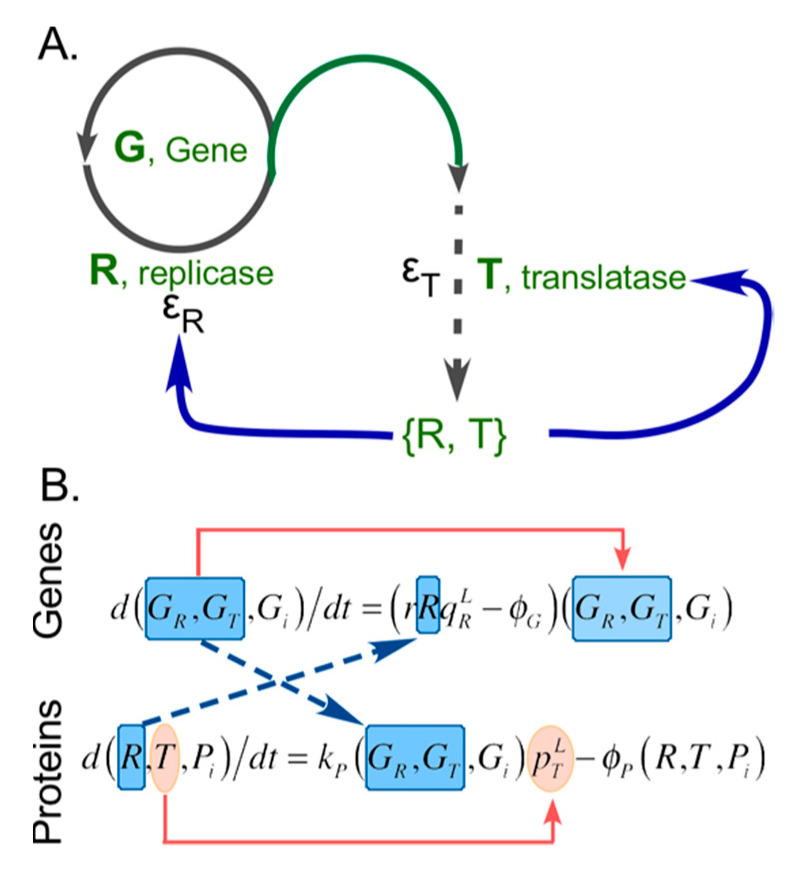
Coupling of replication to translation illustrated by the Gene-Replicase-Translatase model system. (**A**) The rudiments of the GRT system are the replicase and translatase catalysts and their genes (green). The three processes necessary to generate the active catalysts, **R** and **T** are replication (circle), translation (dashed line), and folding (blue lines). No distinction is made between duplication and transcription of the respective genes, **G**, in a world where genetic information is instantiated in RNA. Errors are denoted by ε. (**B**) Differential equations for the two processes [5] are coupled in both directions via the population variables (dashed arrows) and exhibit autocatalysis (red arrows).

**Figure 3 ijms-21-07392-f003:**
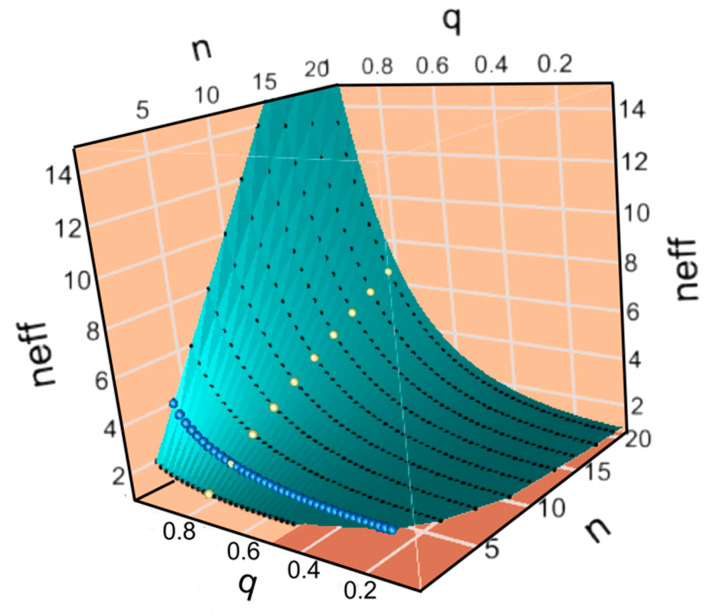
Errors reduce the effective alphabet size, n^eff^, and the information available for functional use, inducing very significant variation over the {n, q} surface. For reference here and in Section 3.3, the alphabet size of nucleic acid bases, n = 4, is marked by blue spheres. The locus of points with q = 0.8 is marked by yellow spheres. Note that n^eff^ approaches the size of the Standard Genetic Code alphabet only when q → 1.

**Figure 4 ijms-21-07392-f004:**
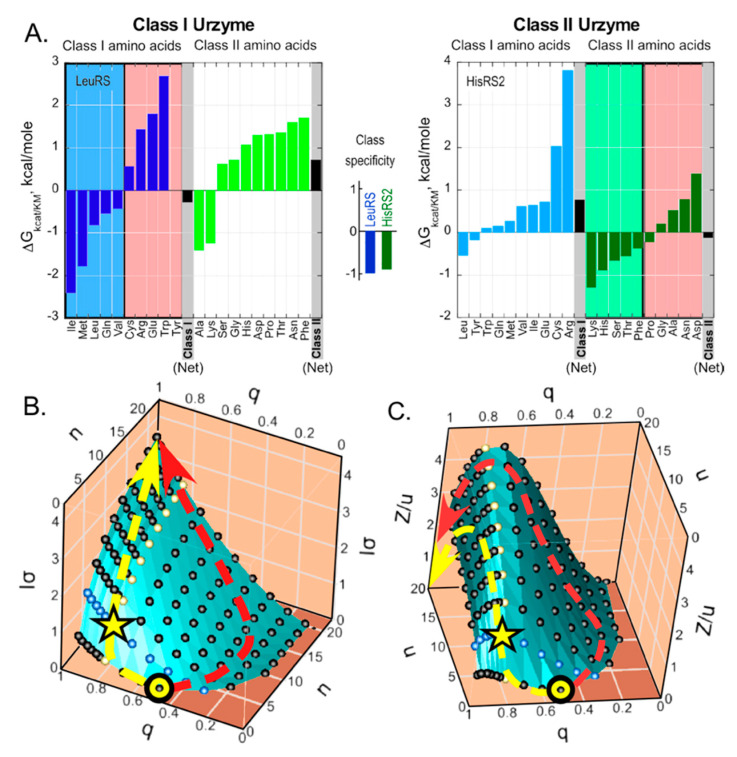
(**A**) Experimental data for comparison with theoretical results derived here. Amino acid specificity spectra for Class I LeuRS and Class II HisRS Urzymes. (**B**) Effects of fidelity (q) and alphabet size (n) on the amount of information transferred per monomer upon synthesis of a polymer from a template (i.e., either replication, transcription, or translation) calculated using the formula for I_σ_, Equation (7). (**C**) Variation of the impedance parameter, Z/u on the {n, q} surface. Points are simulated on the n/q grid using Equation (15). In (**B**,**C**), blue dots are those for a nucleic acid template with four letters; yellow dots indicate the locus of all points with q = 0.8, which represents the ridge over which the impedance parameter in C is maximal for any value of q. Continuous q values are sampled from 0.001 up to 0.999~1.0. Points in the {n,q} plane where q < 1/n have no coding significance, and are darkened to indicate idling behavior. Yellow and red dashed arrows suggest alternate paths. Both paths begin at the yellow circle with a binary alphabet and a minimal q × n value ~ 1.0.

**Figure 5 ijms-21-07392-f005:**
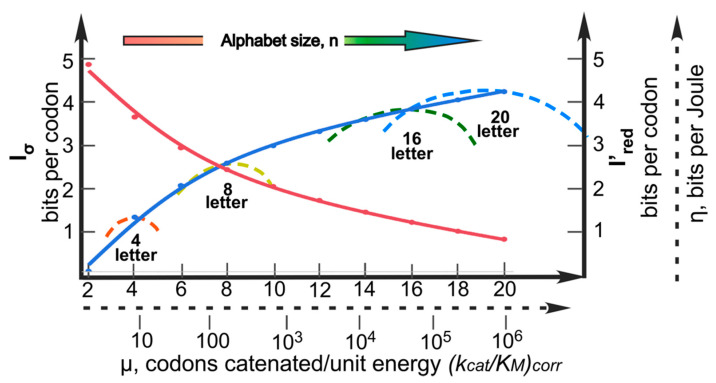
Representative curve (blue) for coded information content I_σ_ (bits per codon or amino acid; left hand Y axis) versus alphabet size, n. Nominal rate of translation, μ, on log(k_cat_/K_M_) is indicated by the dashed proxy scale. Typical curves (dashed, color spectrum) for information transferred per unit energy expended per monomer concatenated, η, (bits per Joule per monomer, arbitrary dashed scale on right hand Y-axis) for aaRS evolution through progressive stages of increasing amino alphabet size, n. Redundancy (red curve), I′_red_ versus n, is averaged out over the codon and amino acid alphabets.

**Table 1 ijms-21-07392-t001:** List of equivalences.

Process	Property	Physical Quantity	Informational Equivalent
**Transfer**	generic entity	energy	information
	reference dimension	time	energy expended in polymer synthesis
	transfer rate	P = IV	η = μI_σ_
	dimensions	energy/time	information/energy
**Flow**	carrier unit	electronic charge	codon or monomer
	carrier flow	I	μ
	dimensions	charge/time	codons/energy
**Potential**	transfer per carrier unit	V	I_σ_
	dimensions	energy/charge	information/codon
**Drive-Flow Linearity**	(Ohm’s law)	V = IR	I_σ_ = μΖ
impedance	transfer potential	dV/dI	dI_σ_/dμ
drive/effort	load	R	Z
flow	driver (internal)	r	z
	maximum	E electromotive force	I_T_ template information
	operating	V = E − Ir	I_σ_ = I_T_ − μz
	operating	I = E/(R + r) = V/R	μ = I_T_/(Z + z) = I_σ_/Z
**Loss During Transfer**	(dissipation)	power loss (heat)	information loss (errors)
		P′ = I(E − V) = I^2^r	η′ = μ(I_T_ − I_σ_) = μ^2^z
low	stalling if R >> r	V → E	I_σ_ → I_T_
		P′ → 0	η′ → 0
		I → 0	μ → 0
high	idle if R << r	P′ → IE	η′ → μI_T_
		V → 0	I_σ_ → 0

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
