# Peer review of "Impedance Matching and the Choice Between Alternative Pathways for the Origin of Genetic Coding"

_ijms, 2020, doi:10.3390/ijms21197392_

Round 1
Reviewer 1 Report
In this article, focusing on the analogy between errors in gene replication and translation, and the impedances in acoustical and electronic energy transducing systems, the authors concluded that the emergence and evolutionary refinement of information transfer in biology could be related to principles identified to govern physical energy flows. For the conclusion, they cited their works on aminoacyl-tRNA synthetase Urzymes, the evolution of which seems to behave according to their theory. By matching the informational impedance associated with the four letter alphabet of genes, they discussed the evolution of the genetic information.
I agree with the mathematical formulations shown in Equations (1)-(11) and also all the figures are plausible. However, my main concern with the article is that Urzymes may not fully function as a concreate example that proves their “impedance theory”. I have to say that there would be a big gap in his logic.
More specifically:
- Why can they assume “four” letter alphabet as the component of the genes in the course of evolution?
- Although they mentioned the editing of misacylated amino acids by aaRSs (Page 3), the editing step is very complicated. The authors should clearly distinguish between just a discrimination by aaRS without editing domain, and editing by aaRS with editing domain. And they should estimate the dissipation of chemical free energy in each step and discuss it based on their “impedance theory”.
- As energy flow, the authors just use the ATP consumption: Hydrolytic expenditure of one ATP is required to attach an amino acid to a tRNA and expenditure of a further two high energy phosphates (as GTP) is required to transfer the amino acid to the nascent peptide at the ribosomal peptidyl transfer centre, hydrolysis of a total of 3 ATP molecules (Page 9). I am afraid that this is oversimplified estimation. Each recognition step of amino acid, ATP and tRNA by aaRS accompany dissipation of chemical free energy and the dissipation of information. How do they handle with them?
- I cannot understand why the average entropic information content of genomic nucleotide sequences can be measured from their GC content (Page 9). Please explain it and also the relationship with their story of the article.
- Their argument “the coincidence of the experimental q value for highly evolved aaRS Urzymes with the blue path argues that the early alphabets required an additional time to maximize fidelity achievable using quasi-species distributions while the coding alphabet size was limiting.” seems to be a kind of forced interpretation. Although the authors may not agree with it, they should at least explain Urzymes a little more in detail.
- Although they wrote “Indeed, it is surmised that the ancestral root Class I and II aaRS enzymes comprised a pair of protein quasi-species that could make only a binary differentiation between the tRNA acceptor stem elements corresponding to the middle nucleotide base of what later became triplet codons, even though that tRNA sequence position could be occupied by any of the four canonical bases.”, both ancestral root Class I and II aaRS enzymes are composed of the mixture of Class I and II amino acids. It seems like a self-contradiction. How can they comment on the fact?
Thus, the authors should carefully focus on more concrete connections between the sufficient experimental and/or theoretical facts on Urzymes and their “impedance theory”. And the author should also be careful to convey their idea appropriately to “general” readers.
Minor points:
Page 16, Line 643
The abbreviations, AARS and CP1, did not seem to be appeared in the text.
Page 19-21, References
There are the mixtures of upper- and lower-case letters in the titles. They should be unified.
Page 20, Line 850
Can the “In Preparation” paper be included in the references?
Page 20, Line 862
“In press” should be “48, 3277-3285”.
Author Response
We apologize for the lengthy incubation period of this revised manuscript. However, we are especially grateful for comments from both reviewers, because they identified several problems that had previously plagued us and motivated us to solve them as definitively as possible. The resulting manuscript is almost entirely re-written as the result of new derivations in §2, which include explicit formulation of the informational impedance parameter, Z, and a new interpretation of previous data on the specificity spectra of experimental aaRS Urzymes from Class I and II. Re-organization of the presentation has shifted speculative paragraphs to the discussion in §3. The result appears to be a much more lasting contribution.
I agree with the mathematical formulations shown in Equations (1)-(11) and also all the figures are plausible. However, my main concern with the article is that Urzymes may not fully function as a concreate example that proves their “impedance theory”. I have to say that there would be a big gap in his logic.
This comment, among others from both referees, moved us to:
(i) reconsider our mathematical framework for the impedance matching analogy and clarify the importance of the mathematical expressions for Iσ, the information transferred in synthesizing a biological polymer of any type; and Z, the impedance parameter; and neff, the effective alphabet size. The latter two quantities were not defined in the previous submission; they clarify the relevance to genetics of effort vs flow relationships in information as well as the clear instantiation of idling and stalling as parallels to the literature on impedance matching in physics.
(ii) re-frame our argument for its relevance to both physics and biology, particular the importance of the sparse available data on the specificity of aaRS Urzymes, and clarify how the redundancy of triplet codons buffers the loss of information at high error rates;
(iii) reorganize the manuscript in considerable detail to clarify the salient results and implications of the mathematical formalism.
More specifically:
- Why can they assume “four” letter alphabet as the component of the genes in the course of evolution?
We had not actually stated that assumption. However, our earlier presentation did obscure the reasons why the four-letter nucleic acid alphabet provides a reference point for coding alphabets of all dimensions, from 2 to >20. Changes in the manuscript to address this confusion are too numerous to cite individually here.
- Although they mentioned the editing of misacylated amino acids by aaRSs (Page 3), the editing step is very complicated. The authors should clearly distinguish between just a discrimination by aaRS without editing domain, and editing by aaRS with editing domain. And they should estimate the dissipation of chemical free energy in each step and discuss it based on their “impedance theory”.
We clarify discussion of the role of aaRS editing domains by returning to the original identification by Pauling of the need for such domains, and cite the work by John Hopfield, who actually measured the energetic cost of inserting an incorrect homolog of isoleucine by IleRS in the third paragraph of §2.2.
- As energy flow, the authors just use the ATP consumption: Hydrolytic expenditure of one ATP is required to attach an amino acid to a tRNA and expenditure of a further two high energy phosphates (as GTP) is required to transfer the amino acid to the nascent peptide at the ribosomal peptidyl transfer centre, hydrolysis of a total of 3 ATP molecules (Page 9). I am afraid that this is oversimplified estimation. Each recognition step of amino acid, ATP and tRNA by aaRS accompany dissipation of chemical free energy and the dissipation of information. How do they handle with them?
A new section, §2.7 delves into this question more deeply than in the original submission.
- I cannot understand why the average entropic information content of genomic nucleotide sequences can be measured from their GC content (Page 9). Please explain it and also the relationship with their story of the article.
- Their argument “the coincidence of the experimental q value for highly evolved aaRS Urzymes with the blue path argues that the early alphabets required an additional time to maximize fidelity achievable using quasi-species distributions while the coding alphabet size was limiting.” seems to be a kind of forced interpretation. Although the authors may not agree with it, they should at least explain Urzymes a little more in detail.
Revised §3.1 includes enhanced treatment of the specificity spectra of Urzymes. Importantly, we now recognize that those spectra contain hints at the specificity values for both a binary code and the all-important 4-letter code that eventually made it possible to craft enzymes with a high degree of specificity. §3.3 details the application of impedance matching to the evolution of the Standard Genetic Code.
- Although they wrote “Indeed, it is surmised that the ancestral root Class I and II aaRS enzymes comprised a pair of protein quasi-species that could make only a binary differentiation between the tRNA acceptor stem elements corresponding to the middle nucleotide base of what later became triplet codons, even though that tRNA sequence position could be occupied by any of the four canonical bases.”, both ancestral root Class I and II aaRS enzymes are composed of the mixture of Class I and II amino acids. It seems like a self-contradiction. How can they comment on the fact?
There is no contradiction between having complementary genes for Class I and II aaRS and having both genes use complementary patterns of two kinds of amino acids. That question is, indeed, an object of current research, and it represents one of the outstanding challenges remaining in order to understand the origin of genetic coding. This question lies outside the scope of this paper, but has been discussed previously in multiple publications. Notably, the active-site residues in Class I enzymes are all activated by Class II aaRS, and conversely.
Thus, the authors should carefully focus on more concrete connections between the sufficient experimental and/or theoretical facts on Urzymes and their “impedance theory”. And the author should also be careful to convey their idea appropriately to “general” readers.
We certainly agree that the previous submission required considerable work to clarify what we had outlined. The manuscript revision attempts to address this concern throughout.
Minor points:
Page 16, Line 643
The abbreviations, AARS and CP1, did not seem to be appeared in the text.
This is no longer the case in the revision, which articulates the connection between the CP1 insertion domains and enhanced specificity in §3.3 lines 631-639.
Page 19-21, References
There are the mixtures of upper- and lower-case letters in the titles. They should be unified.
The variation in titles of cited articles arises from the journals in which they were published.
Page 20, Line 850
Can the “In Preparation” paper be included in the references?
This is an invited review; it has been submitted and publication is expected.
Page 20, Line 862
“In press” should be “48, 3277-3285”.
Corrected in revision
Reviewer 2 Report
Major comments
The authors have developed quantitative relationships necessary to confirm the analogy, based on the previous observation. It is interesting that the authors have developed quantitative relationships between the dissipative losses of both chemical free energy and information and the impedances in acoustical and electronic energy transducing systems.
However, I have to state that there a serious flaw in the manuscript as described below.
- In the case that analogy could be detected between two different systems, it would be important to obtain new knowledge for solving a difficult problem, which could not be solved only by analysis of the one system. Otherwise, the analogy between the two systems should become meaningless for a progress of science. However, it seems to me that the biochemical and genetic problems discussed in the manuscript can be explored independently of the impedances in acoustical and electronic energy transducing systems.
- For example, I guess that Figure 3 can be drawn independently of the physical law, although the physical law was used as reference.
- New biochemical properties or fidelity of gene expression derived from the analogy with the physical law should be presented as summarizing in a Table in the manuscript in order to show that I have misunderstood the content of the manuscript.
Minor comments:
Line 19: “on the one hand” is duplicated as “on the one hand, and, on the other hand,”.
Line 177: in Eq. 1; “k2 and k4” should be “k2 and k4”. That is, “2 and 4” should be written with subscripts as
= {(k2/KM corr )+ (n −1) (k4/KM err )}E0 [M] should be = {(k2/KM corr )+ (n −1) (k4/KM err )}E0 [M]
Line 363: (8-12) should be (8-10)?
Author Response
The authors have developed quantitative relationships necessary to confirm the analogy, based on the previous observation. It is interesting that the authors have developed quantitative relationships between the dissipative losses of both chemical free energy and information and the impedances in acoustical and electronic energy transducing systems.
However, I have to state that there a serious flaw in the manuscript as described below.
- In the case that analogy could be detected between two different systems, it would be important to obtain new knowledge for solving a difficult problem, which could not be solved only by analysis of the one system. Otherwise, the analogy between the two systems should become meaningless for a progress of science. However, it seems to me that the biochemical and genetic problems discussed in the manuscript can be explored independently of the impedances in acoustical and electronic energy transducing systems.
This concern and a similar one voiced by Referee 1 stimulated a complete revision of the derivations of the paper, the organization of the presentation, and the reflections based on experimental data. We believe that this work shines important new light on the evolution of the genetic coding alphabet, and suggest new interpretations of the experimental data discussed in §3.3, and new directions for future experiments.
- For example, I guess that Figure 3 can be drawn independently of the physical law, although the physical law was used as reference.
The graph presented in Figure 3 was entirely determined by the physical analogy developed in the derivation of Equation (6). The same is true of the revised Figure 4B and C, which are plots of the derived formulae for Iσand Z, respectively. The experimental data in Figure 4A have been re-interpreted in light of the insights derived from the plots in B and C.
- New biochemical properties or fidelity of gene expression derived from the analogy with the physical law should be presented as summarizing in a Table in the manuscript in order to show that I have misunderstood the content of the manuscript.
We accept the implicit critique that the submitted manuscript did not clearly outline the new insights derived from the analogy between Ohm’s law and information flow. The re-organization of the manuscript was done, in large part to validate, and to clarify the relevance of such concepts as effort vs flow, idling, and stalling, in information flows associated with biopolymer synthesis. In lieu of the requested table, we include in the introduction an itemized list of our important insights, so that they are evident from the start.
Minor comments:
Line 19: “on the one hand” is duplicated as “on the one hand, and, on the other hand,”.
The revision no longer uses this construction.
Line 177: in Eq. 1; “k2 and k4” should be “k2 and k4”. That is, “2 and 4” should be written with subscripts as
= {(k2/KM corr )+ (n −1) (k4/KM err )}E0 [M] should be = {(k2/KM corr )+ (n −1) (k4/KM err )}E0 [M]
The revision has corrected this problem
Line 363: (8-12) should be (8-10)?
This was a good catch and we are grateful. The equations have been extensively renumbered to include new derivations, but we have tried to ensure that references to the new equations are correct.
Round 2
Reviewer 1 Report
According to the comments, the manuscript has been improved a lot. However, I am wondering why the authors used LeuRS as a representative of Class I aaRSs in Figure 4A. Dr. Carter and his co-workers have extensively worked on Urzymes and they concluded that TrpRS (Class I) and HisRS (Class II) Urzymes may have had modular and functionally active precursors, based on the fact that statistically significant codon middle-base pairing between antiparallel sequence alignments excerpted from TrpRS and HisRS, not LeuRS and HisRS (Mol. Biol. Evol. 30, 1588–1604, 2013). In that sense, the authors should handle TrpRS, not LeuRS, as a representative of Class I aaRSs in Figure 4A. In addition, LeuRS is categorized in Class Ia, but TrpRS is in Class Ic. Furthermore, although TrpRS has the smallest class Ic aaRS with the CP1 insertion of only 74 residues and no editing activity, LeuRS has a long CP1 insertion related editing activity. Are there any inconvenient situations by using TrpRS as a representative of Class I aaRSs? Including them, they should discuss the possibility of TrpRS (Class I) and HisRS (Class II) Urzymes in context of the “Impedance matching”.
Minor point:
Page 25, Line 784
“Figure 43B” should be “Figure 4B”.
Author Response
The reviewer raises an important point that reflects his/her familiarity with several different aspects of our previous work, and which for that reason is appreciated. One of the reasons we are confident that the creation of aaRS Urzymes is a landmark in the analysis of the evolution of genetic coding is that there is such consistency in the experimental and bioinformatics properties of all three of the constructs we have made. A new paragraph introduces §3.1 in which we address the salient points raised by the reviewer and which underscore the validity of our choice to cite the specificity spectra of the LeuRS Urzyme. In fact, the corresponding analysis of the TrpRS spectrum has not been done. The trpRS system has been useful in much more targeted ways that are cited elsewhere at the end of §3.1. However, because limited resources were devoted to different projects over the history of this project, the specificity spectra we have measured are the ones cited in Figure 4A.
Reviewer 2 Report
The authors have appropriately answered my questions and have also extensively revised the previous manuscript, according to my comments and suggestions. So, I consider that the revised manuscript could be accepted.
Author Response
We appreciate the reviewer's decision that the manuscript is acceptable.